# SiGe quantum wells with oscillating Ge concentrations for quantum dot qubits

Thomas McJunkin [1], Benjamin Harpt [1], Yi Feng[1], Merritt P. Losert[1], Rajib Rahman [2], J. P. Dodson[1], M. A. Wolfe[1], D. E. Savage[1], M. G. Lagally[1], S. N. Coppersmith [1,2], Mark Friesen [1] ✉, Robert Joynt [1] & M. A. Eriksson [1] ✉

Large-scale arrays of quantum-dot spin qubits in Si/SiGe quantum wells require large or tunable energy splittings of the valley states associated with degenerate conduction band minima. Existing proposals to deterministically enhance the valley splitting rely on sharp interfaces or modifications in the quantum well barriers that can be difficult to grow. Here, we propose and demonstrate a new heterostructure, the "Wiggle Well", whose key feature is Ge concentration oscillations inside the quantum well. Experimentally, we show that placing Ge in the quantum well does not significantly impact our ability to form and manipulate single-electron quantum dots. We further observe large and widely tunable valley splittings, from 54 to 239 $\mu$eV. Tight-binding calculations, and the tunability of the valley splitting, indicate that these results can mainly be attributed to random concentration fluctuations that are amplified by the presence of Ge alloy in the heterostructure, as opposed to a deterministic enhancement due to the concentration oscillations. Quantitative predictions for several other heterostructures point to the Wiggle Well as a robust method for reliably enhancing the valley splitting in future qubit devices.

Quantum dots formed in silicon-germanium heterostructures are promising candidates for quantum computing, but the degeneracy of the two conduction band minima (or "valleys") in silicon quantum wells can pose a challenge for forming qubits[1–6]. In such structures, the energy splitting between the valley states, $E_v$, is typically tens to a few hundred $\mu$eV and can vary widely due to heterostructure design and unintentional defects[7–19]. The small size and intrinsic variability of $E_v$ has motivated several schemes for modifying or tuning its value. An ambitious scheme to engineer the quantum well barriers, layer-by-layer, has been proposed to increase $E_v$[20,21]. Simpler heterostructure modifications have already been implemented in the laboratory. For example, including additional germanium at the quantum well interface was not found to significantly impact $E_v$[13], while a single spike in germanium concentration within the quantum well was found, theoretically and experimentally, to cause an approximate doubling of $E_v$[17]. Even more practically, $E_v$ can be tuned after device fabrication by changing the applied vertical electric field[14,22,23] or the lateral dot

position[9,15,19], though such tunability tends to be modest in a typical qubit operating range.

Here, we report theory and experiment on a novel Si/SiGe heterostructure, the Wiggle Well, which has an oscillating concentration of germanium inside the quantum well. The wavevector is specially chosen to couple the conduction-band valleys in silicon, thereby increasing $E_v$. This wavevector can be chosen either to couple valleys within a single Brillouin zone or between zones. We measure a quantum dot device fabricated on a Wiggle Well heterostructure grown by chemical vapor deposition (CVD) with Ge concentrations oscillating between 0% and 9%, with wavelength of 1.8 nm, corresponding to the shortest interzone coupling wavevector. The valley splitting is measured using pulsed-gate spectroscopy[24] in a singly occupied quantum dot, obtaining results that are both large and tunable in the range of 54–239 $\mu$eV. We employ an effective mass method to treat Ge concentration variations in the virtual crystal approximation (EMVC) method) to obtain an approximate picture of $E_v$ as a function of the

[1]University of Wisconsin-Madison, Madison, WI 53706, USA. [2]University of New South Wales, Sydney, NSW 2052, Australia.
✉e-mail: friesen@physics.wisc.edu; maeriksson@wisc.edu

oscillation wavelength. We also perform tight-binding simulations of disordered heterostructures using NEMO-3D[25], which qualitatively validates our understanding from the EMVC theory and quantitatively incorporates the effects of both strain and random-alloy disorder. These simulations indicate that the magnitude and range of valley splittings observed in the current experiments can mainly be attributed to natural Ge concentration fluctuations associated with alloy disorder. These theoretical methods are also used to make predictions about a number of additional heterostructures with varying germanium oscillation wavelengths and amplitudes, in which much higher valley splitting enhancements are anticipated.

## Results

We consider a spatially oscillating germanium concentration of the form $\frac{1}{2}n_{Ge}[1 - \cos(qz)]$, as illustrated in Fig. 1(a). Here, $z$ is the heterostructure growth direction, $n_{Ge}$ is the average Ge concentration in the well, and $q$ is the wavevector corresponding to wavelength $\lambda = 2\pi/q$, as indicated in Fig. 1(b). The wavevector $q$ can be chosen to greatly enhance $E_v$. For any Si/SiGe quantum well, the energies of the valley states split in the presence of a sharp interface, but the Wiggle Well produces an additional contribution to $E_v$ due to the oscillating Ge concentration, which gives rise to a potential energy term in the Hamiltonian of the form $V_{osc}(z) \propto [1 - \cos(qz)]$. The electron wavefunctions in the valleys oscillate as $\phi_\pm(z) \propto \exp(\pm ik_0 z)$, where $k_0$ is the location of the conduction band minimum in the first Brillouin zone[3]. Since $k_0$ occurs near the zone boundary, these oscillations are very short-wavelength. For constructive interference that would increase $E_v$, they must be compensated by a corresponding oscillation in the Ge concentration.

Figure 1 (c) shows the Wiggle Well contribution to the valley splitting $E_v(q)$, calculated using the EMVC method for several values of $n_{Ge}$. We observe that the valley splitting is predicted to be enhanced at specific germanium oscillation wavevectors. The wavevector $q \approx 3.5\,nm^{-1}$, corresponding to $\lambda_{long} = 1.8\,nm$, describes coupling between valleys in two neighboring Brillouin zones, as indicated by arrows in the inset. A much larger enhancement of the valley splitting can be achieved for the wavevector $q \approx 20\,nm^{-1}$, corresponding to the much shorter wavelength, $\lambda_{short} = 0.32\,nm$, which describes coupling between the $z$-valley states within a single Brillouin zone, also shown

with arrows. Thus, choosing the oscillation wavelength $\lambda = 2\pi/q$ with care enables the generation of a wavevector in the potential that couples valley minima either between or within Brillouin zones. The large difference in the heights of the two peaks is an extinction effect (destructive interference), caused by a symmetry of the diamond lattice structure. Disorder breaks the symmetry and produces a small peak. The noisy shape of the peak at $q \approx 3.5\,nm^{-1}$ comes from sampling error (see Supplementary Note 1). An additional peak is observed at wavevector $q \approx 10\,nm^{-1}$. We identify this as a harmonic of the taller peak because its height scales as $n_{Ge}^2$, in contrast to the $q \approx 20\,nm^{-1}$ peak, which scales as $n_{Ge}$[26]. At small $q$, there are additional features associated with the details of the barrier interface.

Figure 2 (a) shows a scanning transmission electron micrograph of a Wiggle Well heterostructure grown by chemical vapor deposition (CVD), demonstrating an oscillating concentration of germanium with $\lambda \approx 1.7\,nm$, as described in Methods. Based on this result, the growth parameters were adjusted slightly to achieve the desired $\lambda_{long}$ oscillation period, with an estimated $n_{Ge} = 4.5\%$. The closest match to this value in Fig. 1(c) (red curve) suggests a valley splitting enhancement of about 20 $\mu$eV due to these oscillations. Hall bar devices were fabricated on the heterostructure and measured at a temperature of ~2 K, revealing mobilities in the range of $1\text{-}3 \times 10^4\,cm^2V^{-1}s^{-1}$ for an electron density range of $2\text{-}6 \times 10^{11}\,cm^{-2}$. (See Supplementary Note 2)

To define quantum dots, atomic layer deposition was used to deposit a 5 nm layer of aluminum oxide. Electron beam lithography was used to pattern three layers of overlapping aluminum gates isolated from one another by the plasma-ash enhanced self-oxidation of the aluminum metal, following the procedure described in Ref. 27. (See Methods.) Fig. 2(b) shows a false-colored scanning electron micrograph of a quantum dot device lithographically identical to the one measured. The left half of the device was used for the measurements described below, with a double quantum dot formed in the lower channel and a charge-sensing dot formed in the upper channel. Figure 2(c) shows a stability diagram of the double dot, where the absolute number of electrons can be determined by counting the number of lines crossed in the color plot. All measurements were performed using the last (leftmost) electron transition in this figure, near the magenta star, in a dilution refrigerator with a base temperature below 50 mK.

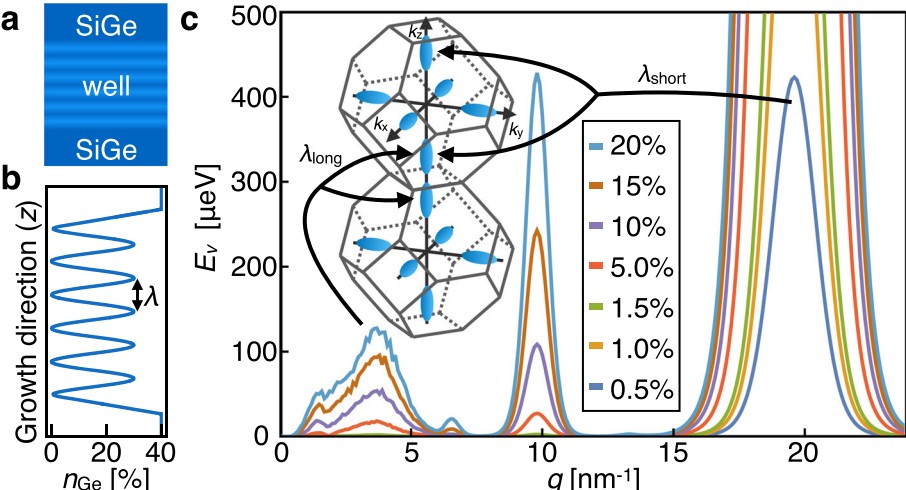

**Fig. 1 | The Wiggle Well. a** Schematic of the Wiggle Well heterostructure, showing Ge oscillations throughout the quantum well. The darker regions have higher Ge concentration. **b** Plot of Ge concentration versus position in a heterostructure with a quantum well with average concentration $n_{Ge}$ of 15% Ge and oscillation wavelength $\lambda$, corresponding to wavevector $q = 2\pi/\lambda$. **c** EMVC predictions for valley splitting contributions ($E_v$ versus $q$) due to Ge concentration oscillations in the quantum well, for $n_{Ge}$ values shown in the inset, and a vertical electric field of

8.5 MV/m. The left inset shows two neighboring Brillouin zones in the silicon conduction band, with constant energy surfaces around the valley minima shown in blue. The peaks at $q \approx 3.5\,nm^{-1}$ arise from Umklapp coupling between the $z$ valleys in neighboring Brillouin zones, and the peaks at $q \approx 20\,nm^{-1}$ arise from coupling between $z$ valleys within a single zone. The peak maxima at $q \approx 20\,nm^{-1}$ lie between 0.4 and 18 meV and are shown on a different scale in Supplementary Fig. 1. Source data are provided as a Source Data file.

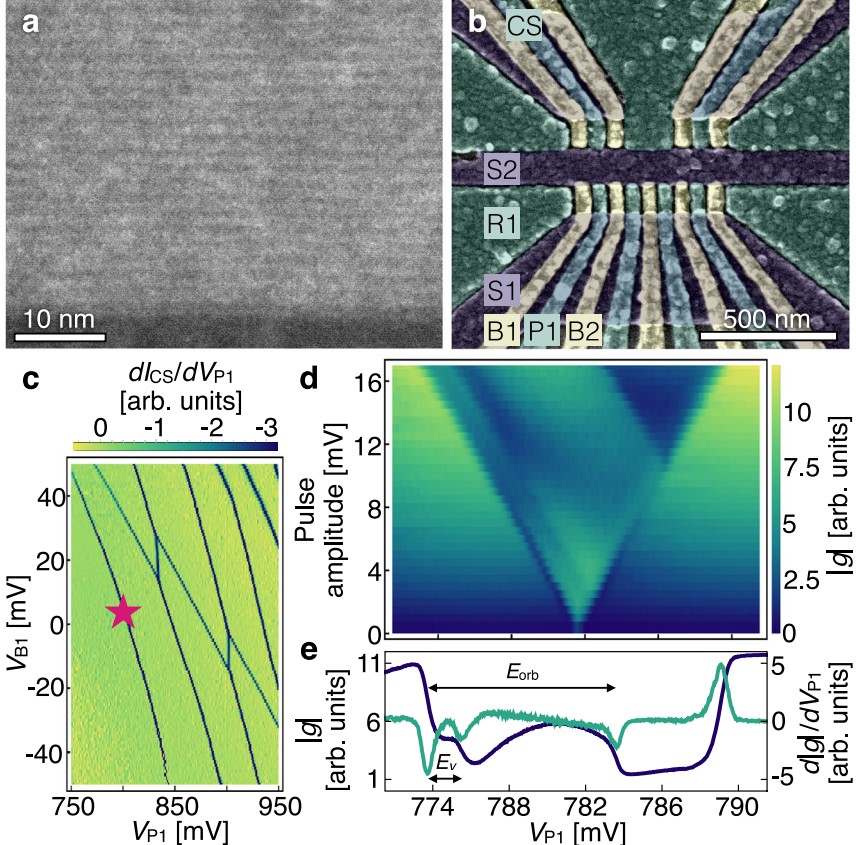

**Fig. 2 | Growth and measurement of a quantum dot device on a Wiggle Well heterostructure. a** High-angle annular dark-field (HAADF) image of a test hetero-structure demonstrating an oscillation wavelength of ~1.7 nm. The lighter regions have higher Ge concentrations. **b** False-color scanning electron micrograph of a quantum dot device lithographically identical to the one measured. The different colors (blue, green, yellow) indicate different gate layers, and relevant gates are labeled. **c** Stability diagram of a quantum dot formed under the leftmost plunger gate in the lower channel, measured using a quantum dot charge sensor in the upper left channel. Here the differential conductance $dI_{CS}/dV_{P1}$ is plotted, where $I_{CS}$ is the current through the charge sensor and $V_{P1}$ and $V_{B1}$ are the voltages on gates P1 and B1, respectively. The dark lines (minima in $dI_{CS}/dV_{P1}$) reveal the voltages at which charge transitions occur in the dots. The measurements presented here are

performed at the last (leftmost) electron transition in this dot, near the magenta star. **d** Pulsed-gate spectroscopy of a singly occupied quantum dot. The dc voltage on gate P1 is swept across the 0-1 electron charging transition while simultaneously applying a square-wave voltage pulse of varying amplitude and 2 kHz frequency, revealing a characteristic V-shape in a lock-in measurement of the transconduc-tance of the charge sensor: $|g| \approx |\delta I_{CS}/\delta V_{P1}|$, where $\delta V_{P1}$ is the pulse amplitude. **e** Extraction of $E_v$ and $E_{orb}$: we repeat 16 P1 voltage scans at the same device tuning as in **d**, for a 16 mV pulse amplitude. The blue curve shows the averaged lock-in response and the green curve shows its derivative with respect to $V_{P1}$. The resulting dips allow us to determine the valley and orbital splittings, $E_v$ and $E_{orb}$, as indicated. Source data are provided as a Source Data file.

The excited-state spectrum of a singly occupied quantum dot was measured using pulsed-gate spectroscopy[19,24,28–31], as shown in Fig. 2(d). Here, the differential conductance of the charge sensor is plotted as a function of the dc voltage on gate P1 vs. the amplitude of the square-wave pulse applied to P1. The data show a sudden change of color when the rate at which electrons enter or leave the dot changes significantly, allowing us to estimate the excited-state energies (see Methods). Figure 2(e) shows in blue the averaged result of 16 individual P1 voltage scans obtained with a 16 mV square-wave amplitude. The green curve is a numerical derivative of the blue curve with respect to $V_{P1}$. Here, the voltage differences corresponding to the valley splitting $E_v$ and the orbital splitting $E_{orb}$ are labeled with arrows. The dips in the differ-entiated signal are fit to extract the voltage splittings, using the methods described in Ref. 19, and then converted into energy splittings using the appropriate lever arm (see Supplementary Note 3), yielding a valley splitting of $164 \pm 3\,\mu eV$ for this particular device tuning.

To develop an understanding of how germanium concentration oscillations and fluctuations can affect the valley splitting, we make use of our ability to change the quantum dot's shape and position in-situ by changing the gate voltages. Importantly, such changes in size and shape can be made while keeping the electron occupation constant.

First, we shift the dot's lateral position by changing the voltages on the screening gates S1 and S2 asymmetrically[19]. Because germanium atoms sit at discrete locations, the concentration oscillations are not identical at all locations in the quantum well; instead, each physical location represents a random instance, which only follows a smooth sine wave pattern when averaged over a wide region. Since the dot is finite in size, changes in position, therefore, cause it to sample local fluctuations of the Ge concentration. Moving the dot in this way also modifies the size and shape of the electron probability distribution in the plane of the quantum well. For this reason, we also perform a second experiment, in which we change the size and shape of the quantum dot while keeping the center position of the dot approximately fixed. In this case, the screening gate voltages S1 and S2 are made more negative, while P1 is made more positive, following the procedure described in Ref. 17, in which the motion of the dot was confirmed through electrostatic modeling.

The orbital and valley splittings resulting from these two different tuning schemes are shown in Fig. 3(a). Both tuning schemes yield a large change in the orbital splitting $E_{orb}$, as shown in the inset to Fig. 3(a), because both change the size and shape of the quantum dot. The valley splitting shows markedly different behavior in the two

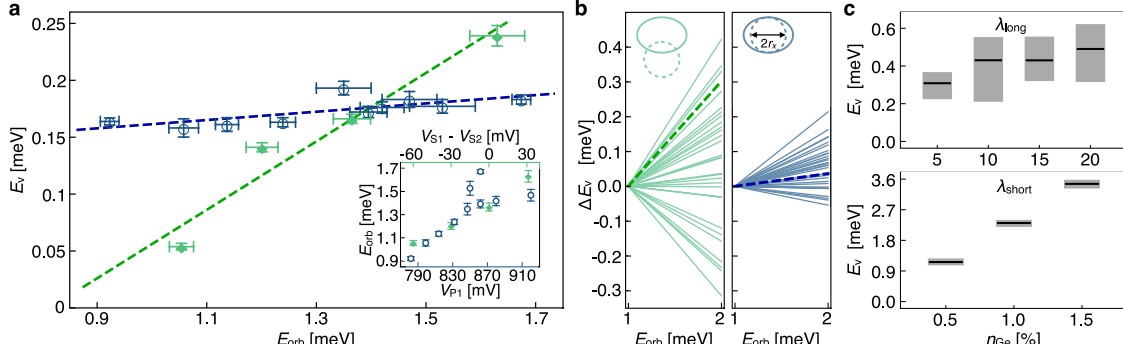

**Fig. 3 | Valley and orbital excitation energies of a Wiggle Well quantum dot.** The voltages applied to the dot are tuned in two ways, both of which change the orbital splitting ($E_{orb}$) substantially but only one of which changes the valley splitting ($E_v$) significantly. Case 1 (filled green diamonds): dot position depends on $E_{orb}$. Case 2 (open blue circles): dot position remains stationary. **a** Inset: Case 1 is achieved by asymmetrically changing voltages on screening gates S1 and S2 (top axis). Case 2 is achieved by changing voltages on S1 and S2 symmetrically, while simultaneously changing the voltage on P1 to compensate (bottom axis). Main panel: valley splittings vary by a factor of 4 for the moving dot, but much less for the stationary dot, over approximately the same range of orbital splittings. Dashed lines are linear fits through the two datasets. Valley splitting is computed by fitting to pairs of spectroscopy peaks [Fig. 2(e)]; error bars reflect the standard error in the peak fits, combined in quadrature, with errors in the lever-arm fits also added in quadrature (see Supplemental Note 3). **b** NEMO-3D tight-binding simulations of Case 1 (left panel) and Case 2 (right panel) scenarios, as depicted by the dot shapes shown in the insets. Simulations include atomistic random-alloy disorder, where the probability of choosing Si or Ge atoms is determined by the Ge concentration profile. Here each curve reflects a unique disorder realization, and we vary the orbital energies (Cases 1 and 2) and dot locations (Case 1 only). Note that $E_v$ values are shifted to align when $E_{orb} = 1$ meV. (Shifted values are labelled $\Delta E_v$.) The dashed lines in **b** are the same as the experimental results in **a**. Here they fall within the statistical range of the randomized simulations, showing consistency with the theory. **c** Statistical sampling of NEMO-3D simulations for several values of germanium concentrations $n_{Ge}$, for $\lambda_{long}$-period (top panel, 40 samples) or $\lambda_{short}$-period (bottom panel, 20 samples) Wiggle Wells. The mean values of the simulations are shown as black lines, along with 25 to 75 percentile ranges (gray bars). Results indicate that alloy disorder plays a dominant role in valley splitting for $\lambda_{long}$ oscillations, with concentration oscillations providing a much smaller enhancement. Source data are provided as a Source Data file.

cases. The first tuning scheme, which moves the dot laterally, to sample different realizations of the Wiggle Well oscillations, yields a large change in the measured valley splitting of nearly 200 $\mu$eV. Here, the variation of $E_v$ is monotonic because the range of motion is similar to the dot radius. The second approach, which does not move the quantum dot, results in a much smaller change in the valley splitting. This large difference in behavior is demonstrated most obviously by the linear fits to the data, which we will compare below to numerical calculations of the valley splitting for many different atomistic realizations of the Wiggle Well. While tunable valley splittings (and closely related singlet-triplet splittings) of Si/SiGe quantum dots have recently been achieved by changing gate voltages[9,14,15,17,19], the observed range of behavior has been modest: for example, 15% tunability with a maximum of $E_v = 213 \mu$eV[15] or 140% tunability with a maximum of $E_v = 87 \mu$eV[19]. Here in contrast, we report a striking > 440% tunability with a maximum of $E_v = 239 \mu$eV.

The EMVC calculations presented in Fig. 1 provide intuition about how oscillating germanium concentrations affect the valley splitting: wave vectors describing the germanium-induced oscillating potential in the quantum well connect valley minima within or between Brillouin zones, as determined by the wavelength of the oscillations. However, such calculations do not provide information about the effect of different atomistic realizations of these oscillations. Moreover, from Fig. 3(a), it is clear that the variations in $E_v$ due to atomistic randomness can be even larger than its mean value.

The strong effect of random alloy disorder on the valley splitting can also be understood from Wiggle Well theory. Due to the finite size of a quantum dot, the electron naturally experiences small layer-by-layer fluctuations of the Ge concentration, as recently explored experimentally[32]. Fourier transforming this distribution assigns random weights across the whole $q$ spectrum in Fig. 1. In particular, weight on the $q \approx 20$ nm peak should have a random but noticeable effect on the valley splitting. In a deterministic Wiggle Well we simply emphasize the weight at certain wave vectors.

To study the competition between deterministic and random enhancements of the valley splitting, we now perform atomistic tight-binding simulations in NEMO-3D using a 20-band sp$^3$d$^5$s* strain-

dependent model[25]. The quantum well concentration profile of Fig. 1(b) is used to construct a heterostructure atom-by-atom, where the probability that an atom is Ge is given by the average Ge concentration at that atom's layer. For all simulations, we assume a typical electric field of 8.5 MV/m.

Figure 3 (b) shows the results of simulations corresponding to the two experiments described in Fig. 3(a). The dots are modeled by the confinement potential $V(x,y) = \frac{1}{2} m_t [\omega_x^2 x^2 + \omega_y^2 (y - y_0)^2]$, where $m_t = 0.19\, m_0$ is the transverse effective mass. In the left-hand panel of Fig. 3(b), the position of the dot ($y_0$) is varied by 20 nm, as consistent with electrostatic simulations reported in Ref. 17. The dot radius along $\hat{x}$ ($r_x$) is also varied, by tuning the orbital energy in the range $\hbar\omega_x = 1$–2 meV, corresponding to $r_x = \sqrt{\hbar/m_t\omega_x} = 14$-20 nm. In the right-hand panel, only $\omega_x$ is varied, over the same range, keeping $y_0$ fixed. In both cases, we choose $\hbar\omega_y = 2$ meV. Each of the curves in Fig. 3(b) is a straight line connecting two simulations. These simulations have different $E_{orb} = \hbar\omega_x$ values, corresponding to 1 or 2 meV, but the same disorder realization. The different curves correspond to different disorder realizations. The left-hand panel confirms that a wide range of valley splittings may be accessed by moving the dot; the experimental slope found for this tuning method (shown by the dashed line) lies within the range of simulation results. The NEMO-3D results in the right-hand panel show a much narrower range of changes in valley splittings, consistent with the experimental observations shown in blue in Fig. 3(a) (dashed line). Here, the center position of the dot does not change, so the dot samples roughly the same disorder for each value of $E_{orb}$. In both panels, $\Delta E_v$ is seen to increase with $E_{orb}$ (on average); this trend can be explained by the prevalence of larger concentration fluctuations in smaller dots, yielding larger valley splittings (on average). These results highlight the ability of random-alloy disorder to affect valley splitting in this system, as compared to the more deterministic concentration oscillations, and the ability of a moving dot to sample these fluctuations.

We now use NEMO-3D tight-binding calculations to make quantitative predictions about valley splitting in other Wiggle Well structures. The top panel in Fig. 3(c) reports results for long-wavelength Wiggle Wells ($\lambda_{long} = 1.8$ nm) with average Ge

concentrations of 5%, 10%, 15%, and 20%. Here, each distribution shows the results of 40 simulations with different realizations of alloy disorder. The bottom panel reports results for short-wavelength Wiggle Wells ($\lambda_{short}$ = 0.32 nm) with average Ge concentrations of 0.5%, 1%, and 1.5%. In this case, results are shown for 20 random-alloy realizations. For all simulations shown in Fig. 3(c), we assume an orbital excitation energy of $\hbar\omega$ = 2 meV. For the long-period Wiggle Well, we see that the effects of alloy disorder are relatively large compared to the deterministic enhancement of the valley splitting caused by Ge oscillations, as indicated by the large spread in results. We also note that the 5% amplitude NEMO-3D results in the top panel are consistent with the experimental valley splittings shown in Fig. 3(a). For the short-period Wiggle Well, NEMO-3D predicts very large boosts in the deterministic contribution to the valley splittings, even for low-amplitude Wiggle Well oscillations.

## Discussion

In summary, we have introduced a new type of silicon/silicon-germanium heterostructure with a periodically oscillating concentration of germanium within the quantum well. Using effective mass theory, we showed that the Wiggle Well can induce couplings between the $z$-valley states, both within a Brillouin zone and between neighboring zones, thereby enhancing the valley splitting. We reported the growth of such a heterostructure with a Ge oscillation period of 1.8 nm within the quantum well, which showed mobility large enough, and corresponding disorder small enough, to form stable and controllable gate-defined quantum dots. Pulsed-gate spectroscopy revealed large valley splittings that were widely tunable through changes in gate voltages. Tight-binding simulations were used to validate the understanding of the experiment and to make predictions about how alloy disorder and structural changes (e.g., in the amplitude and wavelength of the germanium oscillations) can be expected to influence the valley splitting. In the current experiments, simulations indicate that natural Ge concentration fluctuations play a dominant role in determining the magnitude and range of the observed valley splittings. However, the short, 0.32 nm structure is predicted to offer much larger deterministic enhancements of the valley splitting. While this spatial period is short, optimized growth methods have been shown to enable rapid changes in Ge concentrations[33]. For the short-period Wiggle Well, this method should allow 0.93% peak-to-peak Ge concentration oscillations. By further incorporating isotopically purified silicon and germanium into the growth, to suppress hyperfine interactions, the Wiggle Well offers a powerful strategy for improving both coherence times and state preparation and measurement (SPAM) fidelities, by providing reliably high valley splittings.

## Methods

### Theory

We consider a potential that couples the wavefunctions $\phi_{\pm}(\mathbf{r})$ with wavevectors near the valley minima $\mathbf{k} = \pm(0, 0, k_0)$ where $k_0 = 0.84(2\pi/a_0)$ and $a_0 = 0.543$ nm is the lattice constant. The unperturbed wavefunctions are

$$\phi_{\pm}(\mathbf{r}) = \psi(z)e^{\pm ik_0 z}\sum_{\mathbf{K}} c_{\pm}(\mathbf{K})e^{i\mathbf{K}\cdot\mathbf{r}}, \qquad (1)$$

where $\psi$ is an envelope function, the $\mathbf{K}$ are reciprocal lattice vectors, and the $c_{\pm}(\mathbf{K})$ are Fourier expansion coefficients of the cell-periodic part of the Bloch function. The valley splitting $E_v$ induced by the added Ge is[34]

$$\begin{aligned} E_v &= 2\left|\langle\phi_+|V_{osc}(z)|\phi_-\rangle\right| \\ &= 2\left|\sum_{\mathbf{K},\mathbf{K}'} c_+^*(\mathbf{K})c_-(\mathbf{K}')\delta_{K_x,K_x'}\delta_{K_y,K_y'}I(K_z - K_z')\right|, \end{aligned} \qquad (2)$$

where

$$I(K_z - K_z') = \int_{-\infty}^{0} |\psi(z)|^2 e^{iQz}V_0\cos(qz)\,dz, \qquad (3)$$

with $Q = K_z - K_z' - 2k_0$. $|\psi(z)|^2$ is a smooth function with a single peak, so its Fourier transform has a single peak centered at zero wavevector. Hence, $I(K_z - K_z')$ will peak strongly when

$$q = \pm Q = \pm(K_z - K_z' - 2k_0). \qquad (4)$$

Because of the sum over reciprocal lattice vectors in Eq. (2), $E_v(q)$ is expected to be enhanced whenever the condition $K_z - K_z' = \pm(q \pm 2k_0)$ is satisfied. However, a symmetry of the diamond lattice structure leads to a cancellation in the sum over $\mathbf{K}, \mathbf{K}'$ in Eq. (2) when $q = 4\pi/a - 2k_0 = 3.5$ nm$^{-1}$. As described in Supplementary Note 1, the coefficients $c_{\pm}(\mathbf{K})$ in Eqs. (1)-(2) are determined by using the results of a pseudopotential method combined with the virtual crystal method for the disordered SiGe system. This results in a modification of the coefficients that have been previously computed using density functional theory for bulk silicon[34]. The envelope function $\psi(z)$ is found for a quantum well with a vertical electric field of 8.5 MV/m. Further details may be found in Ref. 26.

### Heterostructure growth

The measured heterostructure is grown on a linearly graded SiGe alloy with a final 2 $\mu$m layer of Si$_{0.705}$Ge$_{0.295}$. Prior to heterostructure growth, the SiGe substrate is cleaned and prepared as described in Ref. 13. The substrate is loaded into the growth chamber and flash heated to 825 °C while silane and germane are flowing. The temperature is lowered to 600 °C, at which point a 550 nm 29.5% Ge alloy layer is grown. For the quantum well, the growth begins with a 10 second pulse of pure silane gas at 90 sccm. Then, 90 sccm of silane and 4.88 sccm of germane are introduced for 10.63 s followed by 10 seconds of pure silane. This SiGe–Si pulse sequence is repeated a total of 5 times. The pulse times are tuned to achieve a period of 1.8 nm and a peak Ge concentration of 9%, which was deemed small enough to prevent electrons from leaking out of the quantum well. We note that the actual heterostructure concentration will not achieve a full contrast of 9%, due to atomic diffusion. After the quantum well, a 60 nm Si$_{0.705}$Ge$_{0.295}$ spacer is grown and the heterostructure is capped with a thin 1 nm layer of pure silicon.

### Pulsed-gate spectroscopy

Pulsed-gate spectroscopy is used to measure the valley and orbital splitting of a singly-occupied quantum dot. A square wave voltage is applied to the plunger gate of a dot at a frequency comparable to the tunnel rate to the electron reservoir. The charge sensor current is measured with a lock-in amplifier referenced to the fundamental frequency of the square wave. When the dc voltage of the gate is swept over the dot transition, the electron is loaded and unloaded into the dot as the dot's chemical potential, split by the square wave, straddles the Fermi level of the reservoir. As the amplitude is increased, additional states such as the excited valley state and excited orbital state can be loaded during the high voltage period of the wave, modifying the tunnel rate into the dot. These changes in tunnel rate lead to a changing lock-in response. These changes can be seen in Fig. 2(d).

## Data availability

Raw source data for all relevant figures are available as a 'Source Data' file at https://doi.org/10.5281/zenodo.7374581[35].

## Code availability

The Mathematica files used to generate Fig. 1(c) and Supplementary Fig. 1 are provided as a 'Source Code' file, available at https://doi.org/10.5281/zenodo.7374581[35]. The simulations reported in Fig. 3 and

described in Supplementary Note 4 were performed using NEMO-3D simulation code: https://engineering.purdue.edu/gekcogrp/software-projects/nemo3D/. NEMO-3D is available as open source and is also accessible at nanohub: http://nanohub.org/.

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

## Acknowledgements

We are grateful to A. Saraiva for useful discussions. This research was sponsored in part by the Army Research Office (ARO), through Grant Number W911NF-17-1-0274 (T.M., B.H., Y.F., M.P.L., J.P.D, M.A.W., D.E.S., S.N.C., M.F., R.J., M.A.E.). We acknowledge computational resources and services from the National Computational Infrastructure (NCI) under NCMAS 2021 allocation, supported by the Australian Government (R.R.). Development and maintenance of the growth facilities used for fabricating samples were supported by DOE (DE-FG02-03ER46028). We acknowledge the use of facilities supported by NSF through the UW-Madison MRSEC (DMR-1720415) and the MRI program (DMR-1625348). The views and conclusions contained in this document are those of the authors and should not be interpreted as representing the official policies, either expressed or implied, of the Army Research Office (ARO), or the U.S. Government. The U.S. Government is authorized to reproduce and distribute reprints for Government purposes notwithstanding any copyright notation herein.

## Author contributions

The project was conceived by R.J., T.M., S.N.C., M.F., and M.A.E. T.M. and B.H. fabricated the device and T.M. performed all experiments, with help from B.H., J.P.D., M.A.W., and M.A.E. The heterostructure was grown by D.E.S., with advice from M.G.L. Theoretical calculations were performed by Y.F. and M.L., with advice from R.R., S.N.C., M.F., and R.J. T.M., M.L., S.N.C., M.F., R.J., and M.A.E. wrote the manuscript, with input from all the authors.

## Competing interests

D.E.S., M.F., R.J., S.N.C., and M.A.E. are inventors on U.S. Patent No. 11,133,388, which pertains to a class of heterostructures of which that

studied here is an example. The remaining authors declare no competing interests.
