## [Peer Review File · Nature Communications]

REVIEWER COMMENTS

Reviewer #1 (Remarks to the Author):

It is recognized that in principle Si has the potential of supporting strongly scalable qubit technology, but at the same time it is affected by the extreme sensitivity of the electron wavefunction to the valley physics of the surroundings.

The bottleneck which may prevent the realization of qubit scaling in silicon arises from the combination of having small and variable valley splitting values among the electrons, which in turn are intended to serve as physical basis of the qubits.

The article provides a breakthrough in the valleytronics of the electron wavefunction of individual electrons in silicon, thanks to a new method of tuning the concentration of Ge in SiGe quantum well, consisting of a suitable periodic oscillation of the said concentration, able to heavily increase the valley splitting of hundreds %. The article provides convincing experimental results and a complete model which explains the reasons of the success of the recipe. The physics is fully understood and the idea can bring technology far ahead (and probably could inspire related fields). Thinking to the potential of silicon in the field of quantum computing, this is the most relevant paper I have read in the last five years in the field, as it robustly addresses a major issue which has affected silicon technology up to now. The paper is well conceived and well written. I recommend the publication of the paper as is. Enrico Prati

Reviewer #2 (Remarks to the Author):

The manuscript by McJunkin et al describes an approach to improving valley splitting in Si/SiGe spin qubits using a quantum well structure called the "Wiggle Well". Contrary to published work focusing on sharpening the Si-SiGe interface or reducing the quantum well width, the authors propose a structure with a periodic Ge concentration within the quantum well itself and use various theoretical models to support that such a structure would give rise to large valley splittings. This work is significant in that valley splitting remains a significant impediment to spin qubit operation in the Si/SiGe material systems most notably through state preparation and measurement. It introduces an additional degree of freedom besides spin that enlarges the qubit Hilbert space. Resolving this issue is of high importance and interest within the spin-based quantum computing community, which itself

has seen an explosion in progress over the last few months (as exemplified by recent Nature covers). These facts clearly indicate the topic is of significant interest and worthy of consideration for Nature Communication. Now onto the specifics.

After introducing the concept and theoretical justification, they identify two candidate designs. One long wavelength design with poorer performance and one short wavelength design with excellent modeled performance. However, the high performance design requires very short spacing between Ge concentration peaks within the well and may not be practically realizable (the authors do not comment on this but given the much improved theoretical performance at least some commentary is needed), so they implement the long wavelength design and show a TEM consistent with the target structure. They fabricate quantum dot devices and use them (well, one of them) to probe the achieved valley splitting in the device. They use a measurement technique called pulsed gate spectroscopy (should include a relevant citation here, there are many) to extract the single electron valley splitting from an apparent double-dot device (though one electron is confined below a barrier gate but is not necessarily relevant to the story). This is wise as many people in the field conflate the two-electron singlet-triplet splitting with the valley splitting even though the two-electron singlet-triplet splitting can be reduced by orbital effects.

They perform two types of experiments with this device. By adjusting the biases on adjacent gates, the authors claim they can either keep the electron's position fixed but change its orbital confinement or they can change both parameters at the same time. Conveniently, both types of sweeps sample the same range of orbital energies. Notably, the valley splitting in the fixed position case remains largely unchanged while the moving position case shows a large range of valley splittings (about a 5x range). They then compare these two kinds of experiments to a sophisticated model based on their Si/SiGe heterostructure that can output the valley splitting for a specific alloy instantiation. They then average over these instantiations and show they reproduce the experimental results. This is a little fuzzy to me (because the measurement is not an ensemble but taken from a single device) but the qualitative argument is that the large range of observed valley splittings is consistent with that expected from alloy fluctuations for that amount Ge in the quantum well.

Overall, I think the manuscript is interesting science but there are several important issues I feel need to be addressed prior to publication:

1. A big issue for me is whether or not the authors are claiming to observe a valley splitting enhancement from the periodic structure or not. They don't outright make this claim, but overall the manuscript reads like they want to. From my point of view, such a claim is far from justified by the data. Here are some sub-comments or avenues that would support such a claim or would significantly improve the overall conclusions.

a. The model predicts a strong dependence of the enhancement of valley splitting on the periodicity. The authors show a TEM in Fig 2a of a structure with a slightly smaller than targeted periodicity but don't have any valley splitting data from it. If devices from this structure showed a clear reduction in valley splitting compared to the target structure, this would be a strong justification for claiming the periodicity enhancing valley splitting (which is the primary claim for why such a structure is interesting).

b. The authors show that their structure produces a range of valley splittings beyond the typical tuning range shown in other works for typical quantum wells without intentional Ge in the well. However, in light of other work like Wuetz et al (which shares some of the same authors as this work) this wider tuning range could simply be due to enhanced alloy fluctuations. A nice experiment would be to show that a periodic structure outperforms one with a uniform doping (but same average concentration as the periodic structure). This would be a cool result.

c. All of the modeling assumes that the Ge concentration oscillations are full contrast, but it is very unclear if this is realizable in a CVD growth system or even realized in their structure. Perhaps a line-cut through the TEM in Fig 2a could be shown to state whether or not full contrast is realized. Presumably reductions in contrast would then make a continuous spectrum of results between a periodic structure and a uniform alloy structure.

d. The values of valley splitting achieved in this work are not significantly beyond work published by other groups or even from this same group. This makes it very hard to understand if such a design actually improves valley splitting or not.

e. How many periods of Ge does the actual device have? How was this chosen? How was the target peak concentration chosen? How can the design be optimized?

f. How does such an approach compare to simply narrowing the quantum well without introducing Ge?

2. The authors state that long-wavelength concentration oscillations ($q \ll k_0$) that "negative interference" prevents such a design from coupling the degenerate valleys. It seems like "negative interference" is a poor word choice. Isn't it that $q \ll k_0$ oscillations simply don't matter once the integration over z is carried out? It's more like such oscillations don't contribute to coherent interference of the traveling waves.

3. The authors say that the feature near $q \sim 10 \text{ nm}^{-1}$ is likely a harmonic. Because this is a simulated structure and the dominant feature is $q \sim 20 \text{ nm}^{-1}$ can't they be more definitive than "likely"?

4. The authors state Hall measurements of mobilities without context. Are these mobilities good enough to form qubits? What mobility is too low? Are these mobilities dominated by alloy scattering from the Ge in the well? What should the reader take away from this information?

5. The authors say that they use a double-dot system to perform these measurements, but as far as I can tell all of the measurements are for a single electron. Does the double-dot matter? Is it odd that one of the electrons is confined under the gate designed to control the coupling to the bath?

6. I find the bias dependent experiments and their comparison to modeling very confusing. How are the authors verifying the statement that they fix the location of electron for the first type of sweep? I get that the valley splitting doesn't change much for this and is a possible justification, but it isn't presented that way. If not for the valley splitting remaining nearly unchanged, how is this statement justified?

a. How are these experiments compared to a model? Presumably there is some sense of how much physical distance the electron is being translated, but this is unknown as the only control knob are gate biases? Is there an electrostatic solver being used to understand how much the electron translates for a given bias excursion? How is this captured by the model that predicts the valley splitting?

b. Why is it ok to model the quantum dot's potential as cylindrically symmetric? The images in Fig 3 indicate that the shape of the wavefunction is elliptical. Does this just not matter? Do the authors measure both orbital energies or just one?

c. Why does it make sense to compare the average of the different alloy instantiations in Figure 3b to the experiment in 3a? Perhaps if the data in 3a represented a statistical sampling of valley splittings from the same structure it would make sense, but otherwise it seems profoundly confusing to make this comparison.

7. Why does the distribution of valley splittings broaden non-monotonically in the top panel of figure 3c? It gets very broad for a 10% Ge concentration but then narrows? Is there some intuitive reason for this?

8. The short period Wiggle Well seemingly performs much better in theory, but is it realizable? The authors should at least comment about such a structure.

9. I think the Wiggle Well is a cool idea but am very concerned about alloy scattering reducing mobility and subsequently preventing qubit operation. Does this manuscript offer any insights into whether or not this is an issue?

Again, I think this work is timely and interesting. It addresses a key issue in Si/SiGe spin qubits but it doesn't quite hold together. The experimental data simply doesn't support the most exciting conclusions of the theory, so it is unclear what one should take away from it. One conclusion could be that the periodic structure is interesting in theory, but that the experimental results do not show significant valley splitting improvement over published designs (like a 3nm well) and that it is likely the range of performance is entirely described by alloy fluctuations as opposed to enhancement from the periodicity. I think there is some really interesting things going on here, but significant modifications would be required in order for me to support publication.

Reviewer #3 (Remarks to the Author):

In the manuscript „SiGe quantum well with oscillating Ge concentrations for quantum dot qubits“ by T. McJunkin et al., the authors introduce a new idea to enhance or tune the valley splitting of a Si/SiGe quantum dot. Their innovation is to grow the strained Si quantum-well layer with an oscillating concentration of Ge atoms. Using tight-binding, the valley splitting for different Ge profiles is calculated and compared to measured valley splittings obtained by a well-established method called pulse state spectroscopy. The measured valley splitting can be tuned within an unprecedented large range. The comparison to the predicted values from tight-binding is complicated by alloy disorder within the Ge concentration profile. Therefore, the authors form the quantum dots at various (lateral) positions and various diameters and compare the range of measured valley splitting with the tight-binding calculations for various disorder realizations. As result, the experimental data does not conflict with their tight-binding calculations, but the authors cannot proof (and they do not claim it) that the valley splitting is mainly governed by the oscillating Ge concentration either. More experimental data would be needed for such a claim. In this sense, the inspiring and novel idea of the Wiggle Well is not fully backed up by the experiment. Just the random presence of Ge in the quantum well (discussed in Ref. 31) might lead to similar experimental observations.

Increasing and controlling the valley splitting in Si/SiGe quantum dots is a significant challenge for the scaling up the number of Si/SiGe qubits, the most promising semiconductor based qubit. The innovative approach of the authors to modulate the Ge concentration in the quantum well might justify publication in Nature Communication, if the items listed below are discussed:

Major items:

- 1) Please comment how the alloy disorder is included in the tight-binding calculations in Fig. 3b. Why is ΔE_V monotonic or even linear as a function of E_{orb} , although the alloy disorder is short range by nature? Thus, should not ΔE_V fluctuate as a function of E_{orb} , if alloy disorder dominates in Fig. 3b?
- 2) To what extent is the valley splitting of the Wiggle Well (with λ_{long}) dominated by the ordered (oscillating) Ge concentrations (Fig. 3c) instead of alloy disorder? Comparing to only alloy disorderd Ge concentration in Ref 31 predicts a larger mean value of the valley splitting at 20% Ge concentration.
- 3) Why is E_V as a function of E_{orb} relatively smooth for case 2? Does the short-range nature of alloy disorder imply larger fluctuations of valley splitting for small variations of E_{orb} ?
- 4) In turn, why is E_V as a function E_{orb} monotonically in a wide E_{orb} range? Does it require a special (very lucky) realization of alloy disorder to observe a large tuning range?
- 5) The asymmetry in Fig. 3b case 2 with respect to $\Delta E_V=0$ suggests that not enough disorder realizations are calculated. Or is there a reason, why the extremal negative slopes is smaller than the corresponding positive slope?
- 6) How much of distance (compared to d) is the quantum dot maximally displaced in case 1. The inset of Fig. 3b suggest that the displacement is about the same as d . Is this true?
- 7) How does the out-of-plane electric field E_z change as a function of QD shape and position (Fig. 3). Can a change of E_z already explain a change of E_z in the valley splitting?
- 8) Please comment in the manuscript, whether a Wiggle Well with λ_{short} can be practically grown (by CVD or molecular state epitaxy).
- 9) Why was the lever arm of the V_{P1} not determined by pulse gate spectroscopy from the Zeeman-splitting?
- 10) Why are there very similar valley splittings for λ_{short} and for the Ge concentrations 1.5 % and 5.0 % in Fig. S1? Ge concentration of 10% is then very different from 5 %.
- 11) The authors might mention the necessity to use spin-less Ge nuclei to avoid lowering of spin qubit dephasing time due to hyperfine interaction.

Response to the Referee Comments for the Manuscript *Electronic properties of SiGe quantum wells with oscillating Ge concentrations*

We thank the Referees for providing their reports. We find them to be generally optimistic and very constructive. As detailed below, we have thoroughly addressed all comments and questions.

We would like to point out that, since this manuscript was originally submitted, our theoretical understanding of the Wiggle Well has progressed. In the previous version, an approximate method was used to describe the Bloch functions of SiGe. In the new version, we use a much more rigorous pseudopotential method, taking into account the random alloy disorder present in SiGe. This has resulted in a new theoretical manuscript (arXiv:2206.08331). The results of the new calculations are also incorporated into the present manuscript. Specifically, we have replaced Fig. 1(c), and we have modified (i) the theoretical discussion on the right-hand side of p.2, (ii) the ‘Theory’ section in Methods, and (iii) Supplemental Sec. S1. The most significant change apparent in our new results is that the valley splitting peak associated with λ_{long} in Fig. 1(c) is now substantially smaller than our previous prediction, while the λ_{short} peak is largely unchanged.

A second change we would like to point out is that, in response to comments from two of the Referees (and taking into account the reduced size of the λ_{long} peak), we have changed the focus of the manuscript, from emphasizing the role of Ge concentration oscillations in the quantum well, to the more general role of Ge in the quantum well, and the corresponding physics of alloy disorder. We are very excited about this new interpretation of our experiments, and especially want to emphasize that our new understanding in no way diminishes the importance of the short-period Wiggle Well for future experiments.

A key contribution of our manuscript is to explain intuitively why germanium in the quantum well — even random germanium positions — leads to enhanced valley splitting: calculations of the effects of oscillating germanium concentration, as in Fig. 1 of our manuscript, enable an understanding of valley splitting by considering the Fourier transform of the fluctuating potential energy. Thus, this work connects atomistic tight-binding, which can be quite accurate but difficult to understand intuitively, with physical intuition, through Fig. 1 in the manuscript.

These changes are explained in detail in the following response letter, together with full responses to all Referee comments. With this, we feel the manuscript is very substantially improved, and we look forward to receiving your further correspondence.

In the following letter, we reproduce the Referee comments in full, using blue italic font. Our response to the Referees is shown in black font, and changes to the manuscript are shown in red font.

I. REFEREE 1

Referee Comments: *It is recognized that in principle Si has the potential of supporting strongly scalable qubit technology, but at the same time it is affected by the extreme sensitivity of the electron wavefunction to the valley physics of the surroundings. The bottleneck which may prevent the realization of qubit scaling in silicon arises from the combination of having small and variable valley splitting values among the electrons, which in turn are intended to serve as physical basis of the qubits.*

The article provides a breakthrough in the valleytronics of the electron wavefunction of individual electrons in silicon, thanks to a new method of tuning the concentration of Ge in SiGe quantum well, consisting of a suitable periodic oscillation of the said concentration, able to heavily increase the valley splitting of hundreds %. The article provides convincing experimental results and a complete model which

explains the reasons of the success of the recipe. The physics is fully understood and the idea can bring technology far ahead (and probably could inspired related fields). Thinking to the potential of silicon in the field of quantum computing, this is the most relevant paper I have read in the last five years in the field, as it robustly addresses a major issue which has affected silicon technology up to now. The paper is well conceived and well written. I recommend the publication of the paper as is.

Response: We thank the Referee for kind words and the recommendation to publish.

II. REFEREE 2

Referee Comments: The manuscript by McJunkin et al describes an approach to improving valley splitting in Si/SiGe spin qubits using a quantum well structure called the “Wiggle Well”. Contrary to published work focusing on sharpening the Si-SiGe interface or reducing the quantum well width, the authors propose a structure with a periodic Ge concentration within the quantum well itself and use various theoretical models to support that such a structure would give rise to large valley splittings. This work is significant in that valley splitting remains a significant impediment to spin qubit operation in the Si/SiGe material systems most notably through state preparation and measurement. It introduces an additional degree of freedom besides spin that enlarges the qubit Hilbert space. Resolving this issue is of high important an interest within the spin-based quantum computing community, which itself has seen an explosion in progress over the last few months (as exemplified by recent Nature covers). These facts clearly indicate the topic is of significant interest and worthy of consideration for Nature Communication. Now onto the specifics.

After introducing the concept and theoretical justification, they identify two candidate designs. One long wavelength design with poorer performance and one short wavelength design with excellent modeled performance. However, the high performance design requires very short spacing between Ge concentration peaks within the well and may not practically realizable (the authors do not comment on this but given the much improved theoretical performance at least some commentary is needed), so they implement the long wavelength design and show a TEM consistent with the target structure. They fabricate quantum dot devices and use them (well, one of them) to probe the achieved valley splitting in the device. They use a measurement technique called pulsed gate spectroscopy (should include a relevant citation here, there are many) to extract the single electron valley splitting from an apparent double-dot device (though one electron is confined below a barrier gate but is not necessarily relevant to the story). This is wise as many people in the field conflate the two-electron singlet-triplet splitting with the valley splitting even though the two-electron singlet-triplet splitting can be reduced by orbital effects.

They perform two types of experiments with this device. By adjusting the biases on adjacent gates, the authors claim they can either keep the electron’s position fixed but change its orbital confinement or they can change both parameters at the same time. Conveniently, both types of sweeps sample the same range of orbital energies. Notably, the valley splitting in the fixed position case remains largely unchanged while the moving position case shows a large range of valley splittings (about a 5x range). They then compare these two kinds of experiments to a sophisticated model based on their Si/SiGe heterostructure that can output the valley splitting for a specific alloy instantiation. They then average over these instantiations and show they reproduce the experimental results. This is a little fuzzy to me (because the measurement is not an ensemble but taken from a single device) but the qualitative argument is that the large range of observed valley splittings is consistent with that expected from alloy fluctuations for that amount Ge in the quantum well.

Overall, I think the manuscript is interesting science but there are several important issues I feel need to be addressed prior to publication.

Response: We thank the Referee for their thorough review and for positive comments.

...the high performance design requires very short spacing between Ge concentration peaks within the

well and may not practically realizable (the authors do not comment on this but given the much improved theoretical performance at least some commentary is needed)

We agree with the Referee that such a comment would be helpful. For CVD (the most widely used growth method for Si/SiGe structures), the rate at which Ge concentration can be changed depends on the details of the chemical precursor gases used and the growth temperature. A 2018 paper by Kohen et al. (David Kohen et al 2018 *Semicond. Sci. Technol.* **33**, 104003), reports a method using chlorosilane that produces a compositional rate of change of 5.8% Ge/nm (as reported in Fig. 3b of that paper). For the short wavelength (0.32 nm period) structure we discuss here, this published rate will enable just over 0.93% Ge concentration oscillations peak-to-peak, meaning that this result would produce a structure at nearly the 0.5% line (shown in blue) in Fig. 1(c) in the manuscript. Even better results may of course be possible if the growth is optimized for the oscillatory structure we describe. While the growth system we had available for this work does not support that chemistry, this is indeed an important point and we have added the following discussion of this topic, and a reference to the paper mentioned above, to the final discussion of results on page 5 of the revised manuscript.

The short, 0.32 nm structure has been shown in calculations to offer even larger deterministic enhancements of the valley splitting. While this spatial period is short, optimized growth methods have been shown to enable rapid changes in Ge concentrations [32]. For the short-period Wiggle Well, this method should allow 0.93% peak-to-peak Ge concentration oscillations. By further incorporating isotopically purified silicon and germanium into the growth, to suppress hyperfine interactions, the Wiggle Well offers a powerful strategy for improving both coherence times and state preparation and measurement (SPAM) fidelities, by providing reliably high valley splittings.

...pulsed gate spectroscopy (should include a relevant citation here, there are many)

We now cite the first paper describing pulsed-gate spectroscopy (Elzerman, 2004) on page 1 of the revised manuscript. In addition to this new reference, we have also shifted some other references to where they better describe experimental details. We note that all the references in the original version of the manuscript were retained in the new version.

1. A big issue for me is whether or not the authors are claiming to observe a valley splitting enhancement from the periodic structure or not. They don't outright make this claim, but overall the manuscript reads like they want to. From my point of view, such a claim is far from justified by the data. Here are some sub-comments or avenues that would support such a claim or would significantly improve the overall conclusions.

This is an important issue on which our view has evolved. We are now in complete agreement with the Referee. In the previous version of the manuscript, we used a simple method to account for Ge alloy disorder in the EMVC theory, and to subsequently compute the valley splitting in Fig. 1. That method suggested that the deterministic enhancement of the valley splitting by the Wiggle Well should be similar to the average enhancement caused by alloy disorder, which led to our previously lukewarm claims. In the meantime, however, we have greatly improved our theoretical understanding of Bloch functions in SiGe, resulting in new, improved estimates. These new calculations give somewhat smaller predictions for the valley splitting in the long-period Wiggle Well, due to an extinction effect, which is now discussed in a long new passage shown in red text on p.2 of the manuscript, as well as the ‘Theory’ section of Methods, and Supplemental Sec. S1. We have also written a new manuscript explaining this physics in detail (Ref. 34: arXiv:2206.08331).

Combining these new theory results with the large variability of the valley splitting observed in

Fig. 3a, it is now abundantly clear that the valley splitting enhancement and variability reported in our manuscript should be attributed to alloy disorder. Thus, in the present manuscript, we no longer make claims about the valley splitting being noticeably enhanced by the intentional concentration oscillations.

On the other hand, we believe our experiment is the first to incorporate Ge throughout the quantum well. While the valley splitting we report is not a world record, it is very close. The valley splitting variability we report is, to our knowledge, far and away the largest ever reported. In the current version of our manuscript, we therefore focus more strongly on the physics of alloy disorder, by applying our theoretical understanding of concentration oscillations to explain the large observed valley splitting.

Combined with the fact that the short-period Wiggle Well should still provide a very large, deterministic enhancement of the valley splitting, as confirmed by several complementary theoretical methods, we feel the results presented here are extremely compelling.

This new change of focus is mentioned as follows in the abstract:

Experimentally, we observe large and widely tunable valley splittings, from 54 to 239 μeV in a single-electron dot, **which we attribute to random concentration fluctuations that are amplified by the presence of Ge alloy in the heterostructure.**

Additional changes are found in the middle of p.3:

To develop an understanding of how germanium concentration oscillations **and fluctuations can** affect the valley splitting....

In a new paragraph at the bottom of p.3:

The strong effect of random alloy disorder on the valley splitting can also be understood from Wiggle Well theory. Due to the finite size of a quantum dot, the electron naturally experiences small layer-by-layer fluctuations of the Ge concentration, as recently explored experimentally [31]. Fourier transforming this distribution assigns random weights across the whole q spectrum in Fig. 1. In particular, weight on the $q \approx 20$ nm peak should have a random but noticeable effect on the valley splitting. In a deterministic Wiggle Well we simply emphasize the weight at certain wave vectors.

And at the top of p.4:

To study the **competition between deterministic and random enhancements of the valley splitting**, we now perform atomistic tight-binding simulations in NEMO-3D using a 20-band $\text{sp}^3\text{d}^5\text{s}^*$ strain-dependent model.

a. The model predicts a strong dependence of the enhancement of valley splitting on the periodicity. The authors show a TEM in Fig 2a of a structure with a slightly smaller than targeted periodicity but don't have any valley splitting data from it. If devices from this structure showed a clear reduction in valley splitting compared to the target structure, this would be a strong justification for claiming the periodicity enhancing valley splitting (which is the primary claim for why such a structure is interesting).

As described above, we no longer make claims about deterministic enhancements of the valley splitting in our experiment, so the suggested experiment is no longer needed. However we agree with the Referee that the question about robustness of valley splitting with respect to variations in the oscillation period is important and should be considered in future work.

b. The authors show that their structure produces a range of valley splittings beyond the typical tuning range shown in other works for typical quantum wells without intentional Ge in the well. However, in light of other work like Wuetz et al (which shares some of the same authors as this work) this wider tuning range could simply be due to enhanced alloy fluctuations. A nice experiment would be to show that a periodic structure outperforms one with a uniform doping (but same average concentration as the periodic structure). This would be a cool result.

We agree with the Referee that this would be a very interesting experiment. Since valley splitting enhancements are relatively weak in the long-period Wiggle Well, compared to the effects of alloy disorder, such an experiment would only be meaningful for a short-period Wiggle Well. Indeed, we are planning for such an experiment in the future. However, this is a long-term endeavor. We also note again that such an experiment is no longer needed here, since we do not make claims about deterministic enhancement of the valley splitting.

c. All of the modeling assumes that the Ge concentration oscillations are full contrast, but it is very unclear if this is realizable in a CVD growth system or even realized in their structure. Perhaps a line-cut through the TEM in Fig 2a could be shown to state whether or not full contrast is realized. Presumably reductions in contrast would then make a continuous spectrum of results between a periodic structure and a uniform alloy structure.

We understand and appreciate the Referee's concern. Unfortunately, since TEM images are obtained by averaging over many atoms in a cross-section, they do not provide an accurate measurements of the contrast. The point is well taken though, and we have now added the following sentence to the 'Heterostructure Growth' section of Methods:

We note that the actual heterostructure concentration will not achieve a full contrast of 9%, due to atomic diffusion.

d. The values of valley splitting achieved in this work are not significantly beyond work published by other groups or even from this same group. This makes it very hard to understand if such a design actually improves valley splitting or not.

We believe this comment has been fully addressed in the discussion above, and in the changes to the manuscript, also described above.

e. How many periods of Ge does the actual device have? How was this chosen? How was the target peak concentration chosen? How can the design be optimized?

As discussed in Methods, our quantum well has 5.5 periods of 1.8 nm-period SiGe. This number was chosen to achieve a total well thickness of ~ 10 nm, which is typical for Si/SiGe heterostructures. Since the deposition rate of each layer is linear with time, for a fixed growth temperature, the growth should be reproducible, and nearly the same as the test structure shown in Fig. 2a. The target peak concentration was chosen to provide a noticeable effect (according to our original predictions for enhanced valley splitting), without adding so much Ge that the electrons would leak out of the quantum well. In the future, optimization of the heterostructure could be achieved by establishing a feedback loop involving growth with varying growth parameters, combined with STEM and valley splitting measurements. Indeed, we plan to implement such a feedback loop as we begin to focus on the short-period Wiggle Well. We have made the following change in the 'Heterostructure Growth' section of Methods:

The pulse times are tuned to achieve a period of 1.8 nm and a peak Ge concentration of 9%, **which was**

deemed small enough to prevent electrons from leaking out of the quantum well.

f. How does such an approach compare to simply narrowing the quantum well without introducing Ge?

Narrow quantum wells have been shown to have a significant effect on the magnitude and variability of valley splitting (e.g., see PR Applied **15**, 044033). A fascinating interplay, between deterministic enhancement of the valley splitting and random alloy effects, occurs in such structures due to the significant penetration of the electron wavefunction into the barrier, where alloy effects are strong. A comprehensive description of such behavior is beyond the scope of the current manuscript. However, we are currently preparing a theory manuscript in which this physics is thoroughly studied. We note however that valley splitting enhancements predicted for short-period Wiggle Wells are still much larger than any measurements or predictions in narrow quantum wells.

2. The authors state that long-wavelength concentration oscillations ($q \ll k_0$) that “negative interference” prevents such a design from coupling the degenerate valleys. It seems like “negative interference” is a poor word choice. Isn’t it that $q \ll k_0$ oscillations simply don’t matter once the integration over z is carried out? It’s more like such oscillations don’t contribute to coherent interference of the traveling waves.

These are good arguments, and we agree the issue of interference was not explained clearly in the previous version. However, an important part of the theoretical explanation of Wiggle Well physics does involve destructive interference. The effect is very closely analogous to the suppression of certain X-ray reflections in the Si structure (which of course is due to destructive interference). We have now included an explanation of this “extinction effect” in the new, long text passage on p.2, and in the new theoretical manuscript, Ref. 34. In the same passage on p.2, we also provide a better explanation of the constructive interference offered by the Wiggle Well.

3. The authors say that the feature near $q \sim 10 \text{ nm}^{-1}$ is likely a harmonic. Because this is a simulated structure and the dominant feature is $q \sim 20 \text{ nm}^{-1}$ can’t they be more definitive than “likely”?

We thank the referee for this suggestion. We are able to identify the $q = 10 \text{ nm}^{-1}$ peak as a harmonic because its peak height scales as n_{Ge}^2 , in contrast to the $q = 20 \text{ nm}^{-1}$ peak, for which the peak height scales as n_{Ge} . These scaling arguments are further explained in arXiv:2206.08331. We also include a brief statement about the scaling near the end of the long, new red passage on p.2.

4. The authors state Hall measurements of mobilities without context. Are these mobilities good enough to form qubits? What mobility is too low? Are these mobilities dominated by alloy scattering from the Ge in the well? What should the reader take away from this information?

We thank the Referee for pointing out this omission. We agree that peak mobilities in Si/SiGe are often $\gtrsim 100,000 \text{ cm}^2/\text{Vs}$. However, if one estimates the mean free path of electrons based on our Hall measurements, it is of order $1 \mu\text{m}$, which is much larger than the size of a dot, suggesting that there should be no significant problem with electron localization in our devices. For comparison, we note that systems such as Si-MOS typically have even lower mobilities ($\sim 10,000 \text{ cm}^2/\text{Vs}$), and yet they still report qubit fidelities above 99.9% (e.g., see Nature Electronics **2**, 151). To comment on this issue, we have added the following statement to Supplemental Sec. S2:

The peak mobility reported here is 5-10 times lower than other recently reported values for pure silicon quantum wells [4-6]. However, the estimated electronic mean-free path in this device is $\sim 1 \mu\text{m}$, so we do not expect this mobility to be a limiting factor for qubit formation or performance.

5. The authors say that they use a double-dot system to perform these measurements, but as far as I can tell all of the measurements are for a single electron. Does the double-dot matter? Is it odd that one of the electrons is confined under the gate designed to control the coupling to the bath?

The multi-dot functionality of the device used in this experiment is not important. We show a double-dot stability diagram in Fig. 2 only to demonstrate that the device operates normally, and as expected. However, in our valley splitting measurements, the second dot is fully depleted, and the device operates strictly as a single dot.

In response to the Referee's question about the location of the dot, it is not at all unusual that a dot does not lie directly beneath a plunger gate; such situations can arise due to intentional voltage tunings or may involve disorder-induced electrostatic variations. For the tunings in our experiment, the dot lever arms to the barrier gate B1 and plunger gate P1 are very similar, indicating that it is probably situated between those two gates. Since the chemical potential of the dot and the tunnel rate to the nearby reservoir were found to be 'typical' for this device, we did not attempt to retune the lever arms. It is also useful to note that the gate voltages chosen here play a large role in the dot positions and were chosen specifically to achieve the tunnel rates needed for the spectroscopy we perform.

6. I find the bias dependent experiments and their comparison to modeling very confusing. How are the authors verifying the statement that they fix the location of electron for the first type of sweep? I get that the valley splitting doesn't change much for this and is a possible justification, but it isn't presented that way. If not for the valley splitting remaining nearly unchanged, how is this statement justified?

We apologize for confusion on this point, which was not fully explained in the manuscript. Our claim is not based on the behavior of the valley splitting, but rather on electrostatic modeling in a different experiment, on a device with a gate geometry that was nearly identical to the one used here. The second tuning method (in which the the screening gate voltages are varied symmetrically to keep the dot position fixed while changing the orbital energy and vertical electric field) uses the same procedure as our previous study (Ref. [17]: PRB **104**, 085406). The electrostatics modeling in that study showed that the center of mass of the dot was stationary to within 1 nm.

We have now added a comment about the stability of the dot location at the end of Supplemental Sec. S3, and we have modified the following sentences at the top of p.3 in the main text:

...we also perform a second experiment, in which we change the size and shape of the quantum dot while **keeping the center position of the dot approximately fixed. In this case, the screening gate voltages S1 and S2 are made more negative, while P1 is made more positive, following the procedure described in Ref. [17], in which the motion of the dot was confirmed through electrostatic modeling.**

a. How are these experiments compared to a model? Presumably there is some sense of how much physical distance the electron is being translated, but this is unknown as the only control knob are gate biases? Is there an electrostatic solver being used to understand how much the electron translates for a given bias excursion? How is this captured by the model that predicts the valley splitting?

As noted in our response to the previous question, the lateral motion of the dot was determined through electrostatic modeling in Ref. [17]. For the first tuning method, in which the position of the dot was intentionally varied, the motion was found to be 20 nm, and this information was used in our valley splitting simulations. We have now modified the text as follows on the left-hand column of p.4 to describe this 20 nm motion:

In the left-hand panel of Fig. 3(b), the position of the dot (y_0) is varied by 20 nm, as consistent with electrostatic simulations reported in [17].

b. Why is it ok to model the quantum dot's potential as cylindrically symmetric? The images in Fig 3 indicate that the shape of the wavefunction is elliptical. Does this just not matter? Do the authors measure both orbital energies or just one?

We thank the Referee for pointing out this serious misrepresentation in the text. Only the simulations in Fig. 3c assumed circular dots (for simplicity). The simulations in Fig. 3b assumed dots ranging from ellipsoidal to circular, as consistent with our understanding of the experiments in Fig. 3a. To correct this error, we have modified the main text on p.4 as follows:

Figure 3(b) shows the results of simulations corresponding to the two experiments described in Fig. 3(a). The dots are modeled by the confinement potential $V(x, y) = \frac{1}{2}m_t[\omega_x^2x^2 + \omega_y^2(y - y_0)^2]$, where $m_t = 0.19m_0$ is the transverse effective mass. In the left-hand panel of Fig. 3(b), the position of the dot (y_0) is varied by 20 nm, as consistent with electrostatic simulations reported in [17]. The dot radius along \hat{x} (r_x) is also varied, by tuning the orbital energy in the range $\hbar\omega_x = 1-2$ meV, corresponding to $r_x = \sqrt{\hbar/m_t\omega_x} = 14-20$ nm. In the right-hand panel, only ω_x is varied, over the same range, keeping y_0 fixed. In both cases, we choose $\hbar\omega_y = 2$ meV.

In response to the Referee's other question, we only measure the lowest orbital excitation, corresponding to the long axis of the ellipse (ω_x).

c. Why does it make sense to compare the average of the different alloy instantiations in Figure 3b to the experiment in 3a? Perhaps if the data in 3a represented a statistical sampling of valley splittings from the same structure it would make sense, but otherwise it seems profoundly confusing to make this comparison.

We apologize that the explanation of Fig. 3b in the caption was not clear. The blue and green dashed lines in this panel are only meant to show the experimental results (corresponding to the dashed lines in Fig 3a). The purpose of the figure is to show that the experimental data fall within the statistical range of the randomized simulations. To explain this point, we have modified the following text in the caption:

The dashed lines in (b) are the same as the experimental results in (a). Here they fall within the statistical range of the randomized simulations, showing consistency with the theory.

7. Why does the distribution of valley splittings broaden non-monotonically in the top panel of figure 3c? It gets very broad for a 10% Ge concentration but then narrows? Is there some intuitive reason for this?

This is likely due to the relatively small sample size of the NEMO-3D simulations (40 samples per distribution). Generally, the variance of E_v should increase as the average Ge concentration increases.

8. The short period Wiggle Well seemingly performs much better in theory, but is it realizable? The authors should at least comment about such a structure.

This is the same as the Referee's very first question. In short, we believe that approximately 1% concentration contrasts should be possible. Please see our full response, above.

9. I think the Wiggle Well is a cool idea but am very concerned about alloy scattering reducing mobility and subsequently preventing qubit operation. Does this manuscript offer any insights into whether or not

this is an issue?

This question is the same as the Referee’s Question 4. In short, we do not anticipate the reduced mobility will be harmful for qubit performance. However, it is question of interest that we intend to study in future experiments. Please see our full response, above.

Again, I think this work is timely and interesting. It addresses a key issue in Si/SiGe spin qubits but it doesn’t quite hold together. The experimental data simply doesn’t support the most exciting conclusions of the theory, so it is unclear what one should take away from it. One conclusion could be that the periodic structure is interesting in theory, but that the experimental results do not show significant valley splitting improvement over published designs (like a 3nm well) and that it is likely the range of performance is entirely described by alloy fluctuations as opposed to enhancement from the periodicity. I think there is some really interesting things going on here, but significant modifications would be required in order for me to support publication.

Again, we thank the Referee for these comments, which we largely agree with. We believe the new version of the manuscript fully addresses these issues, and provides an exciting path forward for future experiments.

III. REFEREE 3

Referee Comments: *In the manuscript “SiGe quantum well with oscillating Ge concentrations for quantum dot qubits” by T. McJunkin et al., the authors introduce a new idea to enhance or tune the valley splitting of a Si/SiGe quantum dot. Their innovation is to grow the strained Si quantum-well layer with an oscillating concentration of Ge atoms. Using tight-binding, the valley splitting for different Ge profiles is calculated and compared to measured valley splittings obtained by a well-established method called pulse state spectroscopy. The measured valley splitting can be tuned within an unprecedented large range. The comparison to the predicted values from tight-binding is complicated by alloy disorder within the Ge concentration profile. Therefore, the authors form the quantum dots at various (lateral) positions and various diameters and compare the range of measured valley splitting with the tight-binding calculations for various disorder realizations. As result, the experimental data does not conflict with their tight-binding calculations, but the authors cannot proof (and they do not claim it) that the valley splitting is mainly governed by the oscillating Ge concentration either. More experimental data would be needed for such a claim. In this sense, the inspiring and novel idea of the Wiggle Well is not fully backed up by the experiment. Just the random presence of Ge in the quantum well (discussed in Ref. 31) might lead to similar experimental observations.*

Increasing and controlling the valley splitting in Si/SiGe quantum dots is a significant challenge for the scaling up the number of Si/SiGe qubits, the most promising semiconductor based qubit. The innovative approach of the authors to modulate the Ge concentration in the quantum well might justify publication in Nature Communication, if the items listed below are discussed:

We thank the Referee for their thorough review and positive comments.

1) Please comment how the alloy disorder is included in the tight-binding calculations in Fig. 3b. Why is ΔE_V monotonic or even linear as a function of E_{orb} , although the alloy disorder is short range by nature? Thus, should not ΔE_V fluctuate as a function of E_{orb} , if alloy disorder dominates in Fig. 3b?

In the tight-binding calculations shown in Fig. 3b, we first construct a complete quantum well heterostructure, atom-by-atom, specifying whether a given atom is Si or Ge, as described in the NEMO-3D

discussion on p.4 of the main text, and in the Supplemental Materials. The resulting alloy disorder is explicitly encoded into the random arrangement of atoms in the lattice. This disorder realization is inserted into the tight-binding Hamiltonian, which is then diagonalized to compute the valley splitting. In addition to the disorder realization, there is also a confinement potential that describes the quantum dot: see $V(x, y)$ on p.4 of the main text.

In Fig. 3b, for a given disorder realization, we perform two different simulations. In the left-hand panel, the disorder stays the same, but we change the position of the center of the dot (y_0) as we simultaneously change the dot confinement along one axis ($\hbar\omega_x$). Here, each line connects the solutions for a resulting pair of simulations. Each of these ΔE_v lines is linear because it simply connects two points.

We apologize if the nature of these simulations was unclear in the previous version. To avoid any possible confusion, we have added the following sentence in the middle of p.4:

Each of the curves in Fig. 3(b) is a straight line connecting two simulations. These simulations have different $E_{orb} = \hbar\omega_x$ values, corresponding to 1 or 2 meV, but the same disorder realization. The different curves correspond to different disorder realizations.

Regarding the question of whether ΔE_v should fluctuate as a function of E_{orb} , this is better illustrated in Fig. 3a. In this case, we see that E_v indeed fluctuates as a function of E_{orb} . In Fig. 3b, if more than two simulations were plotted on each curve, we would also observe fluctuations. However since each curve has only two points, they form straight lines.

2) To what extent is the valley splitting of the Wiggle Well (with λ_{long}) dominated by the ordered (oscillating) Ge concentrations (Fig. 3c) instead of alloy disorder? Comparing to only alloy disordered Ge concentration in Ref 31 predicts a larger mean value of the valley splitting at 20% Ge concentration.

The Referee's observation is correct, and is largely consistent with the view of Referee 2. Our response (in brief) is that using our newly improved EMVC theory, which yields somewhat smaller valley splitting estimates for the long-period Wiggle Well, we now feel the observed valley splitting is dominated by alloy disorder. The current experiment is the first (that we are aware of) to intentionally introduce Ge into the quantum well. Despite no evidence for a deterministic enhancement of the valley splitting here, the results are still very exciting because they demonstrate: (1) an average enhancement of the valley splitting, (2) an enormous variability of the valley splitting, which lends itself to tunability, and (3) the possibility of a very large deterministic enhancement of the valley splitting by using the short-period Wiggle Well. This latter result has not been proven experimentally, but it has been confirmed using several complementary theoretical techniques. For a more complete discussion, please our response to Question 1 on p.3, above.

3) Why is E_v as a function of E_{orb} relatively smooth for case 2? Does the short-range nature of alloy disorder imply larger fluctuations of valley splitting for small variations of E_{orb} ?

Here we think the Referee is referring to panel 2 in Fig. 3b, or alternatively, to the blue data in Fig. 3a. This is a very interesting question. For the case where the center position of the dot stays fixed, but the size of the dot changes, the dot wavefunction largely sees the same collection of atoms, even as its size changes. Consequently, the alloy disorder doesn't change significantly and the valley splitting remains approximately constant.

To highlight this point, we have added the following comment to the right-hand column of p.4:

Here, the center position of the dot is unchanged, so the dot samples roughly the same disorder for each

value of E_{orb} .

4) In turn, why is E_V as a function E_{orb} monotonically in a wide E_{orb} range? Does it require a special (very lucky) realization of alloy disorder to observe a large tuning range?

As noted in our previous response, when only the size of the quantum dot changes, the dot largely sees the same collection of atoms as a function of E_{orb} , so the valley splitting remains approximately constant. When the position of the dot also changes, as it does for the case of the green data in Fig. 3a, then the short answer is that valley splitting changes can be both large and nonmonotonic. We have studied this effect extensively (theoretically), and we are currently preparing the results for publication. However we note that when the dot's range of motion is similar to its radius (as it is here), it still sees largely the same disorder over its whole trajectory, so we would expect to observe approximately monotonic behavior of E_v .

We now comment on this behavior in the middle of the right-hand column on p.3:

Here, the variation of E_v is monotonic because the range of motion is similar to the dot radius.

5) The asymmetry in Fig. 3b case 2 with respect to $\Delta E_V = 0$ suggests that not enough disorder realizations are calculated. Or is there a reason, why the extremal negative slopes is smaller than the corresponding positive slope?

This is a good observation. However, we argue that such behavior is actually expected and doesn't necessarily reflect poor statistics. To understand this, we note that increasing the orbital energy reduces the size of the dot and thus the area over which the disorder is sampled. This increases the fluctuations of the Ge concentration and increases the valley splitting (on average), as we have explained in the manuscript. We note that asymmetry is also present in the green data in Fig. 3b; however it is overwhelmed by the effect of the moving dot.

We now comment on this, near the end of p.4:

In both panels, ΔE_v is seen to increase with E_{orb} (on average); this trend can be explained by the prevalence of larger concentration fluctuations in smaller dots, yielding larger valley splittings (on average).

6) How much of distance (compared to d) is the quantum dot maximally displaced in case 1. The inset of Fig. 3b suggest that the displacement is about the same as d . Is this true?

That is correct: the estimated dot motion is about 20 nm, according to measurements and simulations reported in Ref. [17]. The characteristic dot radius is given by $r = \sqrt{\hbar/m_t\omega} \approx 14$ nm for $\hbar\omega = 2$ meV and $r \approx 20$ nm for $\hbar\omega = 1$ meV.

We now clarify these points on p.4:

In the left-hand panel of Fig. 3(b), the position of the dot (y_0) is varied by 20 nm, as consistent with electrostatic simulations reported in [17]. The dot radius along \hat{x} (r_x) is also varied, by tuning the orbital energy in the range $\hbar\omega_x = 1$ -2 meV, corresponding to $r_x = \sqrt{\hbar/m_t\omega_x} = 14$ -20 nm. In the right-hand panel, only ω_x is varied, over the same range, keeping y_0 fixed. In both cases, we choose $\hbar\omega_y = 2$ meV.

We also note that the insets of Fig. 3b have been modified to more correctly represent the simulated geometries.

7) *How does the out-of-plane electric field E_z change as a function of QD shape and position (Fig. 3). Can a change of E_z already explain a change of ΔE_v in the valley splitting?*

Some variations in E_z are expected for different dot tunings; however the enormous range observed in ΔE_v , particularly for case 1, where we move the dot, is inconsistent with E_z variations, which are largely overwhelmed by alloy disorder. As a general rule, E_z cannot be varied significantly, since this causes a change in occupation of the dot. In the NEMO-3D simulations, we simply assume a constant electric field of 0.0085 V/nm.

8) *Please comment in the manuscript, whether a Wiggle Well with λ_{short} can be practically grown (by CVD or molecular state epitaxy).*

This is a good suggestion. The short answer is that we expect that approximately 1% concentration variations should be possible. For a more complete discussion, please see our response above, on page 3 of this response letter, where we explain the modifications we make to the main text, which we also reproduce here:

The short, 0.32 nm structure has been shown in calculations to offer even larger deterministic enhancements of the valley splitting. While this spatial period is short, optimized growth methods have been shown to enable rapid changes in Ge concentrations [32]. For the short-period Wiggle Well, this method should allow 0.93% peak-to-peak Ge concentration oscillations. By further incorporating isotopically purified silicon and germanium into the growth, to suppress hyperfine interactions, the Wiggle Well offers a powerful strategy for improving both coherence times and state preparation and measurement (SPAM) fidelities, by providing reliably high valley splittings.

9) *Why was the lever arm of the V_{P1} not determined by pulse gate spectroscopy from the Zeeman-splitting?*

This is also a good suggestion. However, we are currently preparing a manuscript in which we report a large deviation of the Lande g -factor from 2 in a Wiggle Well. This is a very exciting result, which we want to wait to report. In the current manuscript, we therefore determine the lever arm using the thermal broadening of the electron transition curve, as described in the Supplemental Sec. S3.

10) *Why are there very similar valley splittings for λ_{short} and for the Ge concentrations 1.5% and 5.0% in Fig. S1? Ge concentration of 10% is then very different from 5%.*

This discrepancy was a consequence of the previous approximate method used to calculate Bloch functions in the EMVC theory. In the current theoretical approach, which is much more accurate, the problem no longer exists.

11) *The authors might mention the necessity to use spin-less Ge nuclei to avoid lowering of spin qubit dephasing time due to hyperfine interaction.*

We appreciate the suggestion. It has been added to the concluding discussion in the main text, as follows:

By further incorporating isotopically purified silicon and germanium into the growth, to suppress hyperfine interactions, the Wiggle Well offers a powerful strategy for improving both coherence times and state preparation and measurement (SPAM) fidelities, by providing reliably high valley splittings.

REVIEWER COMMENTS

Reviewer #1 (Remarks to the Author):

I have already expressed the reasons of my positive comments in the previous report. After the amendments, I find the revised manuscript further improved. I recommend the publication as is.
Enrico Prati

Reviewer #2 (Remarks to the Author):

I greatly appreciate the evolution in the understanding of McJunkin and team, likely in response to the comments from myself, referee 3, and other advances in the community (in particular arxiv: 2112.09606 which shares some of the authors of this work). This new insight is significant, and I appreciate the effort the authors put into their response to the reviewer questions. Their responses were very strong and clearly indicated a new understanding of their results. That said, I don't think this was adequately translated to the manuscript itself. This is a challenging result to interpret, especially combined with the emphasis on the wiggle well theory. This is clearly true given the response of Reviewer 1, so the authors should be very careful in how they construct the manuscript.

As it stands, the new manuscript discusses how the measurement results are attributed to random concentration fluctuations but it is entered in an ad-hoc way by adding words or phrases here and there. However, the manuscript is still constructed as

1. We develop a valley splitting theory.
2. We propose a new epitaxial structure to improve valley splitting called the Wiggle Well.
3. We grow a structure based on that proposal and perform TEM to verify we grew our target structure.
4. We perform spectroscopy of the valley splitting for different spatial positions and confinement energies through bias tuning of a quantum dot device in a Will. We observe large values of the valley splitting.
5. We understand those measurements to have valley splittings resulting from random allow fluctuations, which we can also understand from our theory.
6. We use our theory to consider different proposed variations of the Wiggle Well.

Basically, the paper is still constructed as a Wiggle Well validation but with some ad-hoc references to random concentration fluctuations thrown in. For example, the plots in the upper half of Figure 3c should make clear what was stated in the text (the coherent contribution from the long wavelength structure is ~ 20 ueV, so the estimated valley splittings are dominated by concentration fluctuations). The text describing that plot still reads as though the mean valley splitting arises from the deterministic contribution, which is clearly false as that it expected to be only 20 ueV. As it stands, it is still very easy for even an expert reader to come away thinking that the Wiggle Well has been validated (like Referee 1 did) by the measurements. I strongly urge the authors to perform a substantial re-write of the manuscript. All of the work is there, it just needs to be presented clearly so readers can understand what has been shown and what has not been shown. Given the challenges in understanding exactly what the manuscript is attempting to convey, I strongly urge a more substantial re-write.

Reviewer #3 (Remarks to the Author):

The authors very well addressed all items of my review and changed the manuscript and supplements, accordingly. Especially, the item (also raised by another reviewer) about the dominance of alloy disorder discussed in Ref. 32 vs. the controlled enhancement of valley splitting by the modulation of Ge concentration has been clarified in the new version.

As a result, the manuscript contains two outcomes now: First, the idea of enhancement of the valley splitting by a Wiggle-Well (i.e. oscillating Ge concentration). To significantly increase the valley splitting, it requires the short-wavelength of the Ge oscillation of 0.32 nm, which (according to manuscript's outlook) could be grown by CVD. Second, the characterization of the Wiggle-Well version with the long wavelength of 1.8 nm, which shows some improvement in terms of valley splitting and tunability compared to cited Refs. (lines 207 to 208).

The demonstration of growth and measurement of a revolutionary high valley splitting in the short-wavelength Wiggle-Well remains missing in the manuscript. There might be also some new issues arising by the high Ge conc. in these devices besides the hyper-fine interaction (addressed by the authors in the manuscript), such as g-factor variations and enlarged spin-orbit-interaction.

In total, the story of oscillating Ge concentrations for quantum dot qubits has not been finished yet in the presented manuscript. However, in my opinion the idea of the Wiggle-Well is very significant and already justifies publication in Nature Communication, because it suggests a realistic solution for

enhancement of the valley splitting. The enhancement and tunability of valley splitting is the most significant challenge for electron spin-qubits in Si/SiGe, which is the most successful semiconductor system nowadays. That the Ge containing well characterized by the authors already gives an enhancement (within the statistical limits), which is predicted by the author's theory (outcome 2) give a good perspective for the new idea (outcome 1) of the authors.

Minor issues:

Line 73: spelling of "interference"

We are grateful to the Referees for their thorough reviews and helpful suggestions. In this response letter, we address all questions and comments. The Referee comments are reproduced in full using blue font. Our responses are shown in black font. Changes to the manuscript are shown in red font, both here and in the revised manuscript. Please note that the Supplementary Materials are unchanged in this resubmission.

The manuscript is now improved, and we feel it is ready for publication. We look forward to receiving any additional responses.

REFEREE 1

Referee comments: *I have already expressed the reasons of my positive comments in the previous report. After the amendments, I find the revised manuscript further improved. I recommend the publication as is. Enrico Prati*

Response: We thank Dr. Prati for these comments, for his positive recommendation, and for his overall contribution to this work.

REFEREE 2

Referee comments: *I greatly appreciate the evolution in the understanding of McJunkin and team, likely in response to the comments from myself, referee 3, and other advances in the community (in particular arxiv: 2112.09606 which shares some of the authors of this work). This new insight is significant, and I appreciate the effort the authors put into their response to the reviewer questions. Their responses were very strong and clearly indicated a new understanding of their results.*

Response: We thank the Referee for this assessment. Our manuscript has benefitted greatly from previous rounds of Referee reports.

Referee comments: *That said, I don't think this was adequately translated to the manuscript itself. This is a challenging result to interpret, especially combined with the emphasis on the wiggle well theory. This is clearly true given the response of Reviewer 1, so the authors should be very careful in how they construct the manuscript.*

As it stands, the new manuscript discusses how the measurement results are attributed to random concentration fluctuations but it is entered in an ad-hoc way by adding words or phrases here and there. However, the manuscript is still constructed as

- 1. We develop a valley splitting theory.*
- 2. We propose a new epitaxial structure to improve valley splitting called the Wiggle Well.*
- 3. We grow a structure based on that proposal and perform TEM to verify we grew our target structure.*

4. We perform spectroscopy of the valley splitting for different spatial positions and confinement energies through bias tuning of a quantum dot device in a Well. We observe large values of the valley splitting.
5. We understand those measurements to have valley splittings resulting from random alloy fluctuations, which we can also understand from our theory.
6. We use our theory to consider different proposed variations of the Wiggly Well.

Response: We agree with the Referee's summary of our manuscript's construction. Our understanding is that the Referee has no problem with this outline, particularly since point 5 (which was greatly enhanced and improved during the previous round of Referee reports) now clearly and transparently explains our current interpretation of the results, which the Referee also seems to agree with. Indeed, we feel the current construction is the most natural and efficient way to present these results.

Referee comments: *Basically, the paper is still constructed as a Wiggly Well validation but with some ad-hoc references to random concentration fluctuations thrown in. For example, the plots in the upper half of Figure 3c should make clear what was stated in the text (the coherent contribution from the long wavelength structure is ~ 20 eV, so the estimated valley splittings are dominated by concentration fluctuations). The text describing that plot still reads as though the mean valley splitting arises from the deterministic contribution, which is clearly false as that is expected to be only 20 eV. As it stands, it is still very easy for even an expert reader to come away thinking that the Wiggly Well has been validated (like Referee 1 did) by the measurements.*

Response: We disagree with the Referee that our statements in the example (upper half of Fig. 3c) imply any deterministic enhancement of the valley splitting. For example, our final sentence in the caption previously read as follows: "Results indicate that alloy disorder plays a significant role in valley splitting for λ_{long} oscillations." Moreover, Referee 3 also indicates that we have addressed this issue appropriately. However, we accept Referee 2's judgement that the dominant role of alloy disorder over deterministic enhancement could be further emphasized, both in this example and elsewhere in the manuscript.

We have now carefully scoured the manuscript to identify any remaining text that could be viewed as ambiguous in this context, and we have made the following changes:

In the caption for Fig. 3(c):

Results indicate that alloy disorder plays a **dominant** role in valley splitting for λ_{long} oscillations, **with concentration oscillations providing a much smaller enhancement.**

Beginning on line 320 in the conclusions:

In the current experiments, simulations indicate that natural Ge concentration fluctuations play a dominant role in determining the magnitude and range of the

observed valley splittings. However the short, 0.32 nm structure is predicted to offer much larger deterministic enhancements of the valley splitting.

Referee comment: *I strongly urge the authors to perform a substantial re-write of the manuscript. All of the work is there, it just needs to be presented clearly so readers can understand what has been shown and what has not been shown. Given the challenges in understanding exactly what the manuscript is attempting to convey, I strongly urge a more substantial re-write.*

Response: Since the Referee appears to agree with our current outline, we think he/she is asking us to state more clearly what is claimed or not claimed about the Wiggle Well, and asking us to provide such discussion early in the manuscript, to prevent any potential misinterpretations. We are happy to do this. Changes to the manuscript are as follows:

In the abstract, we now clearly distinguish what we have shown experimentally (good quantum dots with large valley splittings, on average), and what is not claimed experimentally, but is predicted theoretically (high valley splittings in other types of Wiggle Well structures):

Experimentally, we show that placing Ge in the quantum well does not significantly impact our ability to form and manipulate single-electron quantum dots. We further observe large and widely tunable valley splittings, from 54 to 239 μeV . Tight-binding calculations indicate that these results can mainly be attributed to random concentration fluctuations that are amplified by the presence of Ge alloy in the heterostructure. Quantitative predictions for several other heterostructures point to the Wiggle Well as a robust method for reliably enhancing the valley splitting in future qubit devices.

At the end of the introduction, beginning on line 53:

These simulations indicate that the magnitude and range of valley splittings observed in the current experiments can mainly be attributed to natural Ge concentration fluctuations associated with alloy disorder. These theoretical methods are also used to make predictions about a number of additional heterostructures with varying germanium oscillation wavelengths and amplitudes, in which much higher valley splitting enhancements are anticipated.

REFEREE 3

Referee comments: *The authors very well addressed all items of my review and changed the manuscript and supplements, accordingly. Especially, the item (also raised by another reviewer) about the dominance of alloy disorder discussed in Ref. 32 vs. the controlled enhancement of valley splitting by the modulation of Ge concentration has been clarified in the new version.*

Response: We thank the Referee for these positive comments, and we note that the Referee disagrees with Referee 2 on this point.

Referee comments: *As a result, the manuscript contains two outcomes now: First, the idea of enhancement of the valley splitting by a Wiggle-Well (i.e. oscillating Ge concentration). To significantly increase the valley splitting, it requires the short-wavelength of the Ge oscillation of 0.32 nm, which (according to manuscript's outlook) could be grown by CVD. Second, the characterization of the Wiggle-Well version with the long wavelength of 1.8 nm, which shows some improvement in terms of valley splitting and tunability compared to cited Refs. (lines 207 to 208).*

The demonstration of growth and measurement of a revolutionary high valley splitting in the short-wavelength Wiggle-Well remains missing in the manuscript. There might be also some new issues arising by the high Ge conc. in these devices besides the hyper-fine interaction (addressed by the authors in the manuscript), such as g-factor variations and enlarged spin-orbit-interaction.

Response: The Referee is correct: g-factors and spin-orbit coupling are both affected and enhanced by the long-period Wiggle Well. We have recently provided a theoretical derivation of this effect in a new preprint, arXiv:2210.01700. This highlights the importance of the current work, and we hope it can be published soon.

Referee comments: *In total, the story of oscillating Ge concentrations for quantum dot qubits has not been finished yet in the presented manuscript. However, in my opinion the idea of the Wiggle-Well is very significant and already justifies publication in Nature Communication, because it suggests a realistic solution for enhancement of the valley splitting. The enhancement and tunability of valley splitting is the most significant challenge for electron spin-qubits in Si/SiGe, which is the most successful semiconductor system nowadays. That the Ge containing well characterized by the authors already gives an enhancement (within the statistical limits), which is predicted by the author's theory (outcome 2) give a good perspective for the new idea (outcome 1) of the authors.*

Response: We are extremely grateful that the Referee shares our enthusiasm for this work.

Referee comment: *Minor issues:
Line 73: spelling of "interference"*

Response: Thank you. The typo has been corrected.

REVIEWERS' COMMENTS

Reviewer #2 (Remarks to the Author):

The manuscript by McJunkin et al seems to have addressed most of the issues I have been highlighting in previous reviews. I think the language has improved the clarity considerably, though I still worry a broad readership will conclude that the Ge oscillations in the Wiggle Well structure gave rise to the improved valley splitting (just as referee 1 concluded after the initial draft), which is clearly not justified. In particular, the abstract references a means to deterministically improve valley splitting and the next sentence claims they "propose and demonstrate" such a structure. However, I argue they only propose such a deterministic structure and instead demonstrate a statistical improvement. I offer as a suggestion that the abstract be modified to say "Tight-binding calculations, and the tunability of the valley splitting, indicate that these results can mainly be attributed to random concentration fluctuations that are amplified by the presence of Ge alloy in the heterostructure as opposed to a deterministic enhancement due to the concentration oscillations." I think it should be made very clear, from the beginning, that the measurements do not indicate that Ge concentration oscillations improved the valley splitting. I prefer this level of clarity, but the editors should feel free to use their own judgement.

It's just a bit tricky because so much effort and text is used to discuss the deterministic enhancement arising from the Ge oscillations (which is a very interesting proposal) but the actual measurements of the demonstrated structure indicate that the results do not arise from that effect but something else (that is also important). It's a difficult balance, but I agree with Referee 3 that the combined themes of the Wiggle Well theoretical proposal and a measured statistical valley splitting enhancement due to concentration fluctuations are high impact to this field and certainly worthy of publication in their own right. The danger is overstating the claims and we should all be wary of drawing unsubstantiated conclusions. I think the authors have taken good steps in this direction, but my preference would be to state it in the abstract. That said, I am in favor of publication and appreciate the time and effort the authors have spent improving the manuscript. The community will certainly reap those rewards.

We are grateful to Referee 2 for helpful suggestions. In this response letter, we address the Referee's final request. Below, Referee comments are reproduced in full using blue font. Our responses are shown in black font. Changes to the manuscript are shown in red font, both here and in the revised manuscript. Please note that the Supplementary Materials are unchanged in this resubmission.

REFEREE 2

Referee comments: *The manuscript by McJunkin et al seems to have addressed most of the issues I have been highlighting in previous reviews. I think the language has improved the clarity considerably, though I still worry a broad readership will conclude that the Ge oscillations in the Wiggle Well structure gave rise to the improved valley splitting (just as referee 1 concluded after the initial draft), which is clearly not justified. In particular, the abstract references a means to deterministically improve valley splitting and the next sentence claims they "propose and demonstrate" such a structure. However, I argue they only propose such a deterministic structure and instead demonstrate a statistical improvement. I offer as a suggestion that the abstract be modified to say "Tight-binding calculations, and the tunability of the valley splitting, indicate that these results can mainly be attributed to random concentration fluctuations that are amplified by the presence of Ge alloy in the heterostructure as opposed to a deterministic enhancement due to the concentration oscillations." I think it should be made very clear, from the beginning, that the measurements do not indicate that Ge concentration oscillations improved the valley splitting. I prefer this level of clarity, but the editors should feel free to use their own judgement.*

It's just a bit tricky because so much effort and text is used to discuss the deterministic enhancement arising from the Ge oscillations (which is a very interesting proposal) but the actual measurements of the demonstrated structure indicate that the results do not arise from that effect but something else (that is also important). It's a difficult balance, but I agree with Referee 3 that the combined themes of the Wiggle Well theoretical proposal and a measured statistical valley splitting enhancement due to concentration fluctuations are high impact to this field and certainly worthy of publication in their own right. The danger is overstating the claims and we should all be wary of drawing unsubstantiated conclusions. I think the authors have taken good steps in this direction, but my preference would be to state it in the abstract. That said, I am in favor of publication and appreciate the time and effort the authors have spent improving the manuscript. The community will certainly reap those rewards.

Response: We thank the Referee for kind words and for helping to improve our manuscript. The Referee asks for one modification in the abstract, which we accept. The modified sentence reads as follows:

Tight-binding calculations, and the tunability of the valley splitting, indicate that these results can mainly be attributed to random concentration fluctuations that are amplified by

the presence of Ge alloy in the heterostructure, as opposed to a deterministic enhancement due to the concentration oscillations.